# Age of Red Cells for Transfusion and Outcomes in Patients with ARDS

**DOI:** 10.3390/jcm11010245

**Published:** 2022-01-04

**Authors:** Jan A. Graw, Victoria Bünger, Lorenz A. Materne, Alexander Krannich, Felix Balzer, Roland C. E. Francis, Axel Pruß, Claudia D. Spies, Wolfgang M. Kuebler, Steffen Weber-Carstens, Mario Menk, Oliver Hunsicker

**Affiliations:** 1Department of Anesthesiology and Operative Intensive Care Medicine (CCM/CVK), Charité-Universitätsmedizin Berlin, Augustenburger Platz 1, 13353 Berlin, Germany; Victoria.Buenger@charite.de (V.B.); lorenz.materne@charite.de (L.A.M.); roland.francis@charite.de (R.C.E.F.); claudia.spies@charite.de (C.D.S.); steffen.weber-carstens@charite.de (S.W.-C.); mario.menk@charite.de (M.M.); oliver.hunsicker@charite.de (O.H.); 2ARDS/ECMO Centrum Charité, Charité-Universitätsmedizin Berlin, 13353 Berlin, Germany; 3Berlin Institute of Health (BIH), Charitéplatz 1, 10117 Berlin, Germany; 4Experimental and Clinical Research Center (ECRC), Charité-Universitätsmedizin Berlin, 10117 Berlin, Germany; alexander.krannich@charite.de; 5Institute of Medical Informatics, Charité-Universitätsmedizin Berlin, 10115 Berlin, Germany; felix.balzer@charite.de; 6Institute of Transfusion Medicine, Charité-Universitätsmedizin Berlin, 10117 Berlin, Germany; axel.pruss@charite.de; 7Institute of Physiology, Charité-Universitätsmedizin Berlin, 10117 Berlin, Germany; wolfgang.kuebler@charite.de

**Keywords:** transfusion, red blood cells, ARDS

## Abstract

Packed red blood cells (PRBCs), stored for prolonged intervals, might contribute to adverse clinical outcomes in critically ill patients. In this study, short-term outcome after transfusion of PRBCs of two storage duration periods was analyzed in patients with Acute Respiratory Distress Syndrome (ARDS). Patients who received transfusions of PRBCs were identified from a cohort of 1044 ARDS patients. Patients were grouped according to the mean storage age of all transfused units. Patients transfused with PRBCs of a mean storage age ≤ 28 days were compared to patients transfused with PRBCs of a mean storage age > 28 days. The primary endpoint was 28-day mortality. Secondary endpoints included failure-free days composites. Two hundred and eighty-three patients were eligible for analysis. Patients in the short-term storage group had similar baseline characteristics and received a similar amount of PRBC units compared with patients in the long-term storage group (five units (IQR, 3–10) vs. four units (2–8), *p =* 0.14). The mean storage age in the short-term storage group was 20 (±5.4) days compared with 32 (±3.1) days in the long-term storage group (mean difference 12 days (95%-CI, 11–13)). There was no difference in 28-day mortality between the short-term storage group compared with the long-term storage group (hazard ratio, 1.36 (95%-CI, 0.84–2.21), *p =* 0.21). While there were no differences in ventilator-free, sedation-free, and vasopressor-free days composites, patients in the long-term storage group compared with patients in the short-term storage group had a 75% lower chance for successful weaning from renal replacement therapy (RRT) within 28 days after ARDS onset (subdistribution hazard ratio, 0.24 (95%-CI, 0.1–0.55), *p <* 0.001). Further analysis indicated that even a single PRBC unit stored for more than 28 days decreased the chance for successful weaning from RRT. Prolonged storage of PRBCs was not associated with a higher mortality in adults with ARDS. However, transfusion of long-term stored PRBCs was associated with prolonged dependence of RRT in critically ill patients with an ARDS.

## 1. Introduction

Transfusion of packed red blood cells (PRBCs) is a frequent and often lifesaving therapy in critically ill patients. In the US, leukoreduced PRBCs can be stored for up to 42 days [1]. During the storage period, erythrocytes undergo complex morphologic and biochemical changes. Altered red cell membrane configuration and plasticity decrease erythrocyte stability, rendering RBCs prone to hemolysis with consecutive liberation of potassium, lactate dehydrogenase, microparticles, hemoglobin, and subsequently heme and iron into the storage solution or the vasculature of the transfusion recipient [2,3]. Besides disturbances of serum electrolytes, the released content of the RBCs causes vascular nitric oxide depletion, vasoconstriction, platelet aggregation, inflammation, mitochondrial damage, and production of reactive oxygen species [3,4,5,6,7]. Furthermore, bioactive lipids, cytokines, and chemokines are generated within the PRBC units during storage and can trigger immune reactions as well as systemic organ injury upon transfusion [8,9].

Whether transfusion of PRBCs after a prolonged storage interval has an impact on patient outcome is debated controversially. While preclinical data and smaller observational clinical studies suggest an association between the duration of PRBC storage and morbidity and mortality in critically ill patients, four large randomized controlled trials (RCTs) failed to detect a difference in survival or adverse effects in Intensive Care Unit (ICU) patients transfused with PRBCs of different storage ages [1,10,11,12,13,14,15,16,17]. The average storage age of the transfused PRBCs differed depending on the trial, from 6 to 13 days for the short-term group, and from 20 to 24 days for the long-term group [14,15,16,17]. Therefore, all four RCTs had insufficient power to examine adverse effects of PRBCs stored for longer than four weeks, the storage interval associated with the maximum of hemolysis [2,3,18,19]. In contrast, smaller, prospective clinical trials focusing on selected study populations demonstrated an increased rate of complications and adverse effects in patients transfused with PRBCs stored for a period closer to the maximum allowed 42 days of storage [20,21,22]. Despite the absence of RCT evidence to avoid transfusion of PRBCs at the end of the 6-week storage period, several countries, including the UK, Ireland, New Zealand, and the Netherlands, have limited the storage period of PRBCs to 35 days or less [17,23].

Acute Respiratory Distress Syndrome (ARDS) is a common complication in ICU patients and is associated with a high mortality of 30–40% [24]. In patients with ARDS, frequent transfusions of PRBCs are necessary to maintain a sufficient blood oxygen carrying capacity to secure the supply of vital organs [25]. In this study, we compared the short-term outcome of patients with ARDS receiving PRBCs with median storage ages shorter or longer than 28 days.

## 2. Materials and Methods

### 2.1. Study Design and Setting

This retrospective cohort study included patients with ARDS admitted to the tertiary ARDS referral center at the Department of Anaesthesiology and Intensive Care Medicine, Charité-Universitätsmedizin Berlin, Campus Virchow-Klinikum, Berlin, who received transfusions of PRBCs within 14 days after ARDS onset. Patients were selected from the electronic patient data management systems during the period from January 2007 to December 2018. Patients transfused with PRBCs of a mean storage age > 28 days were compared to patients transfused with PRBCs of a mean storage age ≤ 28 days for short-term outcome. Patients were transfused with leukoreduced, allogeneic PRBCs. The PRBC additive solutions used by the manufactures during the study period were Adsol, PAGGS-M (phosphate-adenine-glucose-guanosine-saline-mannitol), or SAGM (saline-adenine-glucose-mannitol). The study was approved by the ethical committee of Charité-Universitätsmedizin Berlin (No. EA1/018/19).

### 2.2. Participants

Patients were eligible if they fulfilled the criteria of the Berlin Definition for ARDS [26]. Patients were excluded from analyses if they (1) received extracorporeal life support (ECLS), (2) did not receive PRBC transfusions or had missing storage data, (3) were not transfused during the first 14 days after ARDS onset, (4) received at least one PRBC unit with a storage age > 42 days, or (5) received irradiated PRBCs. Although the majority of manufacturers adhere to the maximum storage of 42 days in the clinical routine, PRBCs produced with PAGGS-M additive solution are approved for storage up to 49 days in Germany. Therefore, to adjust the study cohort to international standards, patients transfused with PRBCs that were stored for >42 days (*n* = 17) were excluded from the analysis.

### 2.3. PRBC Storage Preprocessing and Grouping

Investigating the storage age of PRBCs in non-randomized clinical studies is complex because patients who receive more than one unit of PRBCs are generally transfused with PRBCs of different storage ages. To handle multiple intra-patient storage ages, the mean storage age was calculated for each patient using all PRBC units transfused within the first 14 days after onset of ARDS. The first 14 days after ARDS onset were chosen because the daily transfusion requirements during ARDS treatment are highest during this period, as reported previously [25]. During storage, PRBCs undergo a significant number of biochemical and structural alterations with significant storage-induced damage after 28 days [3,27]. Therefore, a storage age of 28 days was chosen as a cutoff to group patients into a short-term storage group (PRBC mean storage age ≤ 28 days) and a long-term storage group (PRBC mean storage age > 28 days).

In the event of a significant association between the storage groups and at least one study endpoint, a complementary secondary analysis was performed to evaluate whether the transfusion of even a single long-term stored PRBC unit may affect patient outcome. For this, patients were regrouped according to the storage age of the oldest PRBC unit they had received using the same cutoff at a storage age of 28 days.

### 2.4. Endpoints and Data Sources

The primary endpoint was mortality within 28 days after ARDS onset. Secondary endpoints were “failure-free days” composites such as ventilator-free (VFDs), sedation-free, renal replacement therapy-free (RRT), and vasopressor-free days. The definition and analysis of the “failure-free days” composites were based on recent re-appraisal of the composite outcome measures in critical care research [28]. Details are provided in the Appendix A.

As reported previously, the study data were obtained from the hospital’s electronic patient data management systems [25]. Further details are available in the Appendix A.

### 2.5. Bias Handling

Allocation of allogeneic PRBCs with different storage ages is largely independent of medical circumstances. However, grouping patients according to their mean storage duration might induce a selection bias of baseline characteristics. Therefore, patients’ characteristics that were different between the storage groups were identified and considered in multivariable models.

### 2.6. Statistical Analyses

*T*-test and the exact Mann–Whitney U test were used to compare continuous data, as appropriate. Frequencies were tested using Fisher’s exact test. Histograms with normal distribution curves were used to visualize the distribution of mean storage age in the short-term and long-term storage group.

For the primary endpoint of 28-day mortality, the hazard ratio (HR) from Cox proportional hazards regression was used to compare the short-term and long-term storage groups. A multivariable regression model was performed to control for between-group differences, as appropriate. Scaled Schoenfeld residuals and HR-plots were used to check for the proportional hazard assumption. An equal distribution of censoring was evaluated. Kaplan–Meier curves were used to visualize cumulative mortality over the 28-day period.

For the secondary endpoints, a competing risk regression was used retaining (not censoring) patients experiencing the competing event (death) in the risk set [28]. A subdistribution hazard ratio (SHR) was provided that estimated the primary “net effect” size, which is the chance of the long-term storage group compared with the short-term storage group undergoing a prespecified event, such as weaning from mechanical ventilation, sedation, RRT, and vasopressors, accounting for the existence of the alternative outcome of death. Cumulative incidence curves were constructed for each secondary endpoint. All analyses were considered to be non-confirmatory, and a post hoc power analysis was omitted [29]. A two-tailed *p*-value < 0.05 was considered statistically significant. R software, version 3.6.1 (R Project for Statistical Computing) was used for all analyses.

## 3. Results

A total of 1044 patients met the inclusion criteria within the analyzed time period. Of those, 761 patients were excluded because they did not receive PRBC transfusions or had missing storage data, received the first PRBC transfusion later than 14 days after ARDS onset, received at least one PRBC unit with a storage age > 42 days, or received irradiated PRBCs (Figure 1). Thus, a total of 283 patients with storage data of 1954 transfused PRBC units (median five units (IQR, 2–9) per patient) were included for further analysis.

Baseline characteristics of the short-term and long-term storage groups are shown in Table 1. In general, patients received a lung-protective ventilation with low tidal volumes, high PEEP, and low driving pressures. Furthermore, most patients received ARDS rescue therapies such as prone positioning and therapy with inhaled nitric oxide. There were no differences in baseline characteristics such as demographics, ARDS severity, ARDS etiology, or ventilation parameters except for a slightly higher tidal volume in the short-term storage group.

### 3.1. Transfusion Characteristics

Patients in the short-term storage group had a similar median hemoglobin concentration at ARDS onset (10.3 g/dL (IQR, 9.2–11.8) vs. 10.2 g/dL (9.3–12.0), *p* = 0.65), were transfused at a similar median transfusion threshold (8.2 g/dL (IQR, 7.6–9.2) vs. 8.2 g/dL (7.6–8.7), *p* = 0.17), and received a similar median amount of PRBC units (five units (IQR, 3–10) vs. four units (2–8), *p =* 0.14) compared with patients in the long-term storage group (Figure 2, Table 2). The median time from ARDS onset to first PRBC transfusion was not different between the short-term and long-term storage group (16 h (IQR, 12–19) vs. 16 h (13–19), *p* = 0.65). The mean storage age in the short-term storage group was 20 (±5.4) days compared with 32 (±3.1) days in the long-term storage group, resulting in a mean difference of 12 days (95%-CI, 11–13)). The distribution of the storage age in both groups is shown in Figure 2E.

### 3.2. Endpoints

There was no significant difference in 28-day mortality between the short-term storage group and the long-term storage group (HR, 1.36 (95%-CI, 0.84–2.21), *p =* 0.21) (Figure 3). Adjustment for tidal volume did not substantially alter this result (adjusted HR, 1.29 (95%-CI, 0.72–2.31), *p* = 0.38). The median observation time was 20 days (IQR, 17–24) in the short-term storage group and 17 days (15–24) in the long-term storage group. There was no difference in censoring between the two groups (*p* = 0.79) (Appendix A). Within 28 days, 22.6% (95%-CI, 17.1–29.3) of the patients in the short-term storage group compared with 29.2% (20.3–39.9) of the patients in the long-term storage group had died.

Ventilator-free (SHR, 0.79 (95%-CI, 0.49–1.28), *p* = 0.35), sedation-free (SHR, 0.91 (0.65–1.26), *p* = 0.55), and vasopressor-free (SHR, 0.74 (0.52–1.06), *p* = 0.10) days composites were similar between the short-term and long-term storage group (Figure 4A,B,D). In contrast, patients in the short-term storage group had a significantly lower chance of successful weaning from RRT within 28 days after ARDS onset compared to patients in the long-term storage group (SHR, 0.24 (95%-CI, 0.1–0.55), *p <* 0.001) (Figure 4C). Adjustment for preexisting chronic kidney disease confirmed this result (adjusted SHR, 0.24 (95%-CI, 0.1–0.56), *p <* 0.001). There was no difference in censoring between the two groups (*p* = 0.89) (Appendix A). Within 28 days, RRT was successfully weaned in 41.6% (95%-CI, 32.8–51.0) of the patients in the short-term storage group compared with 12.0% (4.9–25.0) of the patients in the long-term storage group.

### 3.3. Secondary Analyses

In a further analysis, patients were regrouped according to the storage age of the oldest PRBC unit they had received to evaluate whether transfusion of even a single long-term stored PRBC unit was associated with a lower chance of successful weaning from RRT. Patient characteristics, transfusion characteristics, and distribution of the storage age of the regrouped patients are available in Appendix A. Besides the higher number of PRBC units transfused in the long-term storage group compared to the short-term storage group (5 (IQR, 3–9) vs. 3 (2–5), *p* < 0.001), there were no differences between the two groups. Patients that had received at least one unit of PRBCs with a storage age > 28 days had a significantly lower chance for successful weaning from RRT within 28 days after ARDS onset compared to patients who only received PRBCs with a storage age ≤ 28 days (SHR, 0.50 (95%-CI, 0.3–0.85), *p =* 0.01) (Appendix A). Adjustment for the number of transfused PRBC units confirmed this result (adjusted SHR, 0.52 (95%-CI, 0.29–0.91), *p =* 0.01). Within 28 days, RRT was successfully weaned in 45.0% (95%-CI, 32.3–58.3) of the patients who only received PRBCs with a storage age ≤ 28 days compared with 26.3% (18.6–35.7) of the patients that had received at least one unit of PRBCs with a storage age > 28 days.

## 4. Discussion

In critically ill patients with ARDS, transfusion of PRBCs with a mean storage age greater than 28 days was not associated with a higher mortality compared to patients transfused with PRBCs with a mean storage age less than 28 days. However, transfusion with PRBCs stored for longer than 28 days was associated with a 75% lower chance for successful weaning from RRT during the first four weeks after ARDS onset. Further analysis indicated that even a single PRBC unit stored for more than 28 days decreased the chance for successful weaning from RRT.

With regard to the primary endpoint, the findings of this study are in concordance with data of previous large RCTs that suggested that the storage age of PRBCs does not impact mortality in Adult ICU patients [14,15,16,17]. Similar data were obtained from RCTs on critically ill pediatric patients [30,31]. However, these studies were designed to test the hypothesis that transfusion of fresher PRBCs would be superior compared to transfusion of standard-issue PRBCs and all four RCTs had insufficient power to examine adverse effects of PRBCs stored for longer than 28 days [17,23]. Several prior observational studies reporting increased mortality among patients transfused with older PRBCs compared storage groups with a mean storage age of transfused PRBCs that was significantly higher in both study arms compared with the study arms of the RCTs [23]. However, unlike a series of mechanistic preclinical studies, most clinical studies to date did not study the extremes of PRBCs storage time with regard to mortality endpoints, i.e., the comparison of transfusion of very fresh PRBCs compared to transfusion of PRBCs at the end of the 6-week storage period [12,32,33]. In this study, outcome after transfusion with standard-issue PRBCs was compared with the outcome after transfusion with PRBCs of a mean storage age > 28 days, the period known to be associated with the storage lesion [3,27].

Increased concentrations of cell-free hemoglobin (CFH) occur in the transfusion recipient after transfusion of PRBCs that have been stored for prolonged intervals [18,21]. CFH is cleared in the kidney, rendering this organ susceptible to hemolysis-associated adverse effects of transfusion with longer-stored PRBCs [34,35]. Glomerular filtration of CFH might further contribute to tubular injury and overall renal damage that is associated with the underlying disease and patients comorbidities [35]. In line with these conceptual pathomechanisms of renal injury caused by transfusion of aged PRBCs, ARDS patients transfused with PRBCs of a mean storage age > 28 days showed a lower chance for successful weaning from RRT compared to patients transfused with PRBCs of the shorter storage group, while both groups did not differ in baseline characteristics and preexisting comorbidities [36]. In concordance, the risk for AKI was higher in patients after orthotopic liver transplantation when longer stored PRBCs were transfused compared to shorter stored PRBCs [37]. In another retrospective study, Koch and coauthors detected a higher incidence of AKI in cardiac surgery patients after transfusion of longer stored PRBCs [13]. However, in both studies, the PRBC storage age discriminated at 14 days. Therefore, both studies essentially compared transfusion with fresh versus standard-issue PRBCs instead of comparing of transfusion with standard-issue PRBCs compared to PRBCs close to the end of the storage period. Furthermore, the increased incidence of AKI detected in the retrospective study of Koch and colleagues was not confirmed in the following RCT of Steiner and colleagues [13,15]. In contrast, the present study demonstrates that even transfusion of a single unit of PRBCs suffices to cause AKI or delay in renal recovery.

Having found a signal for RRT-free days composites in association with the mean storage age of transfused PRBCs, we regrouped the patients according to the storage age of the oldest PRBC unit they had received. Thereby, we could demonstrate that transfusion of a single long-term stored PRBC unit was also associated with a lower chance for successful weaning from RRT within 28 days after ARDS onset. In this analysis, the mean storage ages of both transfusion groups discriminated significantly with 14.6 days. Because baseline characteristics of the patients in the “only short-term” and “at least one long-term” storage group were similar, an additional regression analysis was omitted. A higher number of transfused units in the long-term storage group was expected because chances to receive a unit of long-term stored PRBCs increase with the number of transfusions.

This study has several limitations. Patients with ARDS and treatment with ECMO were excluded from the study because ECMO is known to cause additional hemolysis, an effect that might be confused with the adverse effects attributed to the transfusion of PRBCs with a long storage age [38]. Although the sickest of patients with ARDS were thereby excluded, more than two-thirds of ARDS patients in both study arms still suffered from severe ARDS. Each patient was grouped according to the mean storage age of transfused PRBCs to account for multiple PRBC transfusions of different storage ages. A potential selection bias of baseline characteristics was addressed by use of multivariable models. Due to the retrospective design of this study, no causative conclusions can be drawn from the current observations.

## 5. Conclusions

In critically ill patients with ARDS, transfusion of PRBCs with a mean storage age greater than 28 days was not associated with an increase in 28-day mortality. However, transfusion of long-term stored PRBCs was associated with a lower chance for successful weaning from RRT within 28 days after ARDS onset.

## Figures and Tables

**Figure 1 jcm-11-00245-f001:**
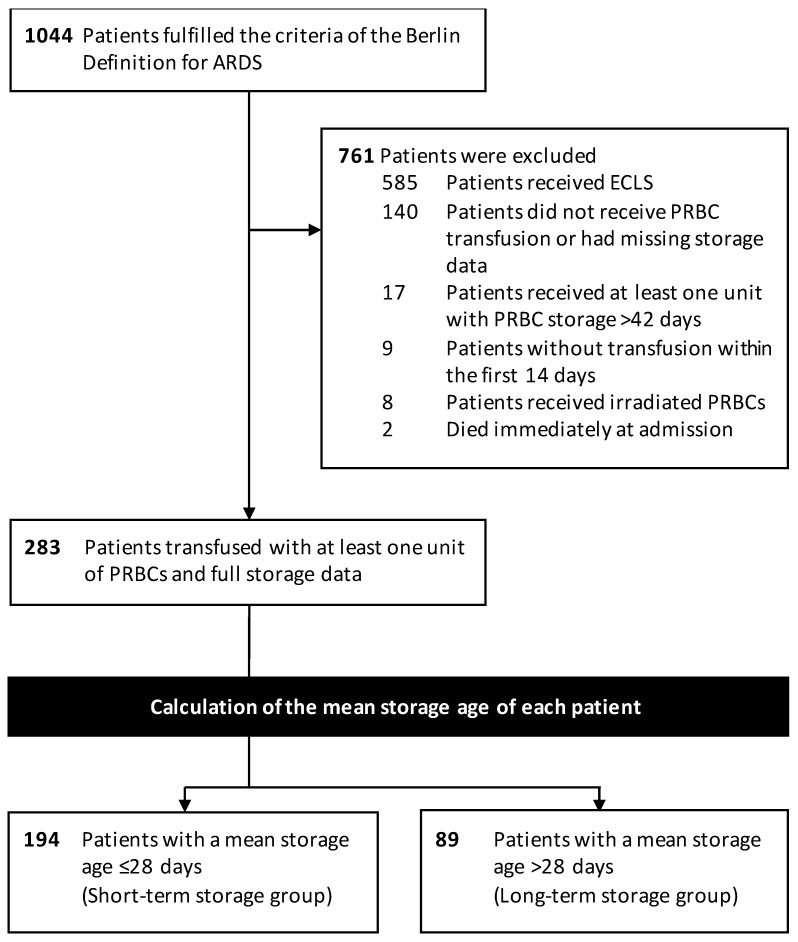
Study Flow diagram.

**Figure 2 jcm-11-00245-f002:**
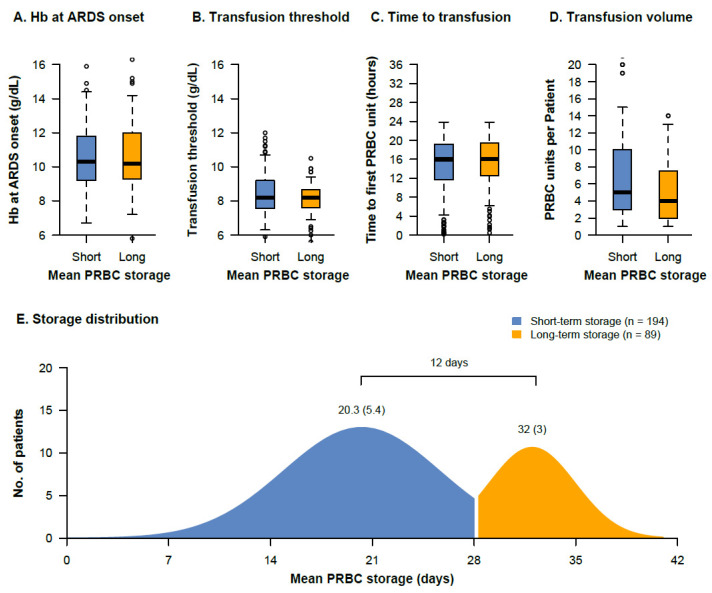
Transfusion characteristics of the short-term and long-term storage groups. The hemoglobin concentrations at ARDS onset (**A**), the transfusion threshold (pre-transfusion hemoglobin) (**B**), the time from ARDS onset to transfusion of the first PRBC unit (**C**), the number of transfused PRBC units within 14 days of ARDS therapy (**D**), and the distribution of the storage age in both groups (**E**) are presented. The mean storage (SD) in each group and the mean difference between the groups is indicated.

**Figure 3 jcm-11-00245-f003:**
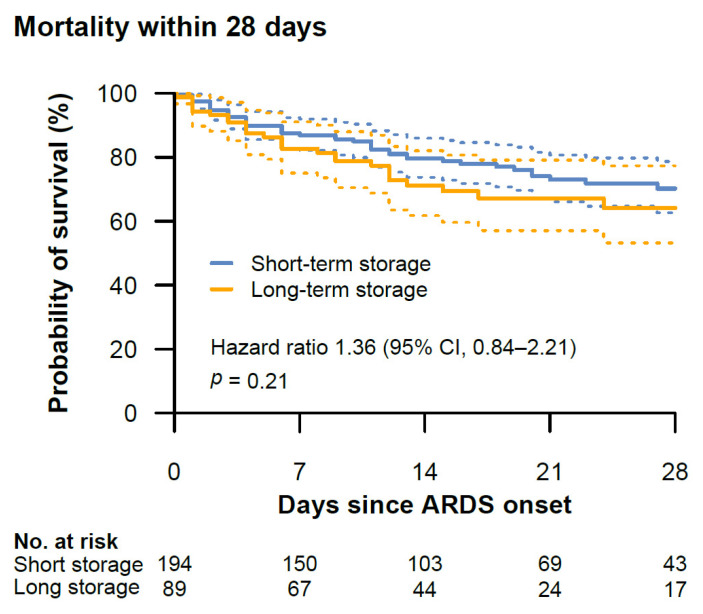
Kaplan–Meier survival curves with 95% confidence intervals (dotted lines) of the mortality within 28 days after onset of ARDS between the short-term and long-term storage group. The hazard ratio is provided with 95% confidence intervals. The median observation time was 20 days (IQR, 17–24) in the short-term storage group and 17 days (15–24) in the long-term storage group. The distribution of censoring was similar between the two groups (*p* = 0.79).

**Figure 4 jcm-11-00245-f004:**
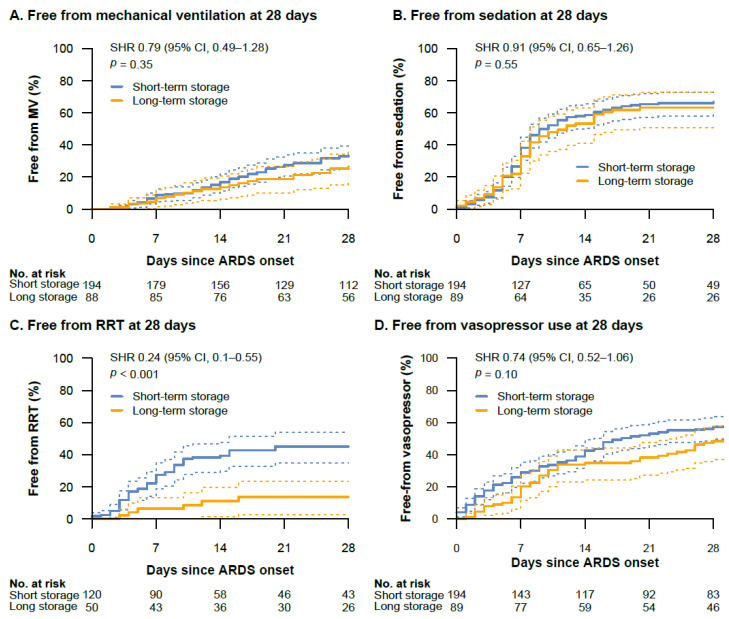
Cumulative incidence curves with 95% confidence intervals (dotted lines) of ventilator-free (**A**), sedation-free (**B**), RRT-free (**C**), and vasopressor-free (**D**) days composites between the short-term and long-term storage groups. The subdistribution hazard ratio (SHR) is provided with 95% confidence intervals. The SHR of the RRT-free days composite was adjusted for the presence of chronic kidney disease. One patient in the long-term storage group did not receive invasive mechanical ventilation during ARDS treatment and was therefore not included in the analysis of the ventilator-free days composite. Definition of abbreviations: MV = mechanical ventilation.

**Table 1 jcm-11-00245-t001:** Characteristics of the patients.

Characteristic	Short-Term Storage Group (*n* = 194)	Long-Term Storage Group (*n* = 89)	*p*-Value
Age (years)	55.50 (42.25, 66.00)	57.00 (47.00, 69.00)	0.248
Male sex, *n* (%)	120 (61.9)	56 (62.9)	0.968
Body mass index (kg/cm)	27.78 (24.39, 32.65)	26.73 (23.67, 31.37)	0.212
Charlson comorbidity index	3.00 (1.00, 5.00)	3.00 (1.00, 6.00)	0.430
Chronic kidney disease, *n* (%)	19 (9.8)	16 (18.0)	0.081
Immunocompromised, *n* (%)	44 (22.7)	22 (24.7)	0.822
SOFA at ARDS onset	11.00 (8.25, 14.00)	12.00 (8.00, 14.00)	0.669
SAPS II at ARDS onset	50.00 (38.00, 64.00)	55.00 (38.00, 70.00)	0.320
RASS at ARDS onset	−5.00 (−5.00, −4.00)	−5.00 (−5.00, −4.00)	0.719
Chronic lung disease, *n* (%)	39 (20.1)	24 (27.0)	0.256
Mechanical ventilation before admission (days)	1.00 (0.00, 4.00)	1.00 (0.00, 4.00)	0.523
ARDS severity, *n* (%)			0.618
Mild	10 (5.2)	7 (7.9)	
Moderate	47 (24.2)	19 (21.3)	
Severe	137 (70.6)	63 (70.8)	
ARDS etiology, *n* (%)			0.495
Pneumonia	102 (52.6)	51 (57.3)	
Aspiration	44 (22.7)	17 (19.1)	
Sepsis	15 (7.7)	11 (12.4)	
Pancreatitis	6 (3.1)	2 (2.2)	
Other	27 (13.9)	8 (9.0)	
Ventilation parameters after initial optimization			
PaO_2_:FiO_2_ (mmHg)	157.37 (109.17, 214.44)	164.47 (111.25, 228.05)	0.401
Oxygenation index	15.19 (9.56, 22.57)	14.35 (8.88, 21.36)	0.244
PEEP (cm H_2_O)	16.60 (14.00, 20.00)	16.00 (13.25, 18.02)	0.075
Driving pressure (cm H_2_O)	15.52 (13.00, 18.00)	15.29 (11.85, 18.84)	0.771
Tidal volume (mL/kg PBW)	6.86 (5.74, 7.76)	6.07 (5.48, 7.05)	0.016
Respiratory rate (breaths/min)	20.00 (18.00, 24.00)	21.00 (20.00, 25.00)	0.306
Compliance (mL/cm H_2_O)	36.40 (27.45, 47.15)	33.85 (26.98, 49.30)	0.735
Rescue therapy			
Inhaled nitric oxide, *n* (%)	131 (67.5)	57 (64.0)	0.660
Prone positioning, *n* (%)	132 (68.0)	52 (58.4)	0.150
Septic shock, *n* (%)	83 (43.2)	37 (42.5)	0.999
Lactate (mg/dL)	16.00 (10.00, 30.00)	16.00 (10.00, 31.00)	0.742
RRT, *n* (%)	120 (61.9)	50 (56.2)	0.439

Definition of abbreviations: SOFA = Sequential Organ Failure Assessment, SAPS = Simplified Acute Physiology Score, RASS = Richmond Agitation-Sedation Scale, PEEP = Positive End-Expiratory Pressure, RRT = Renal replacement therapy. Data are expressed as median (25%, 75% quartiles) or frequencies (%), as appropriate.

**Table 2 jcm-11-00245-t002:** Transfusion characteristics of the patients.

Characteristic	Short-Term Storage Group (*n* = 194)	Long-Term Storage Group (*n* = 89)	*p*-Value
Hemoglobin at ARDS onset (g/dL)	10.30 (9.22, 11.80)	10.20 (9.30, 12.00)	0.647
Transfusion threshold * (g/dL)	8.22 (7.59, 9.20)	8.20 (7.60, 8.65)	0.171
PRBC units transfused per patient with the first 14 days (number)	5.00 (3.00, 9.75)	4.00 (2.00, 8.00)	0.143
PRBC units transfused per patient with the first 28 days (number)	6.00 (2.25, 10.00)	4.00 (2.00, 9.50)	0.193
Mean PRBC storage age per patient (days)	20.23 (5.38)	32.26 (3.10)	<0.001
Oldest PRBC unit per patient (days)	27.81 (8.13)	37.49 (3.31)	<0.001
Time to first PRBC transfusion (hours)	15.90 (11.63, 19.20)	16.08 (12.83, 19.40)	0.657
Patients receiving transfusion of other blood components, *n* (%)			
Platelets	137 (70.6)	57 (64.0)	0.333
Fresh frozen plasma	53 (27.3)	22 (24.7)	0.753

Definition of abbreviations: PRBC = packed red blood cells. Data are expressed as mean (SD), median (25%, 75% quartiles) or frequencies (%), as appropriate. * An individual hemoglobin threshold for RBC transfusion was calculated for each patient as reported previously [25].

## Data Availability

Data are available from the corresponding author on reasonable request.

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
