# Peer review of "Age of Red Cells for Transfusion and Outcomes in Patients with ARDS"

_jcm, 2022, doi:10.3390/jcm11010245_

Round 1

Reviewer 1 Report

Thank you for the opportunity to review this manuscript. It is reassuring to see that there were no mortality differences between the groups, in line with reports from randomized controlled trials. It is curious that only the weaning from RRT showed a difference. As the authors mention, causative conclusions cannot be drawn from this retrospective study.

I recommending a few other clinical trials to the references: 

  • Fergusson DA, Hébert P, Hogan DL, et al. Effect of fresh red blood cell transfusions on clinical outcomes in premature, very low-birth-weight infants: The ARIPI randomized trial. JAMA 2012;308:1443-51.
    - Dhabangi A, Ainomugisha B, Cserti-Gazdewich C, et al. Effect of transfusion of Red Blood Cells with longer vs shorter storage duration on elevated blood lactate levels in children with severe anemia. JAMA 2015;314:2514-23.
    - Spinella PC, Tucci M, Fergusson DA, Lacroix J, Hébert PC, et al. Effect of Fresh vs Standard-issue Red Blood Cell Transfusions on Multiple Organ Dysfunction Syndrome in Critically Ill Pediatric Patients: A Randomized Clinical Trial. JAMA 2019 Dec 10;322(22):2179-2190. 
    - Shah A, Brunskill SJ, Desborough MJ, Doree C, Trivella M, Stanworth SJ. Transfusion of red blood cells stored for shorter versus longer duration for all conditions. Cochrane Database Syst Rev. 2018 Dec 22;12(12):CD010801.

Their incorporation may affect some of the introduction content. For ethical reasons, the clinical trials could not randomize patients to a long-storage RBC transfusion regimen and instead had to compare fresh units to standard units. This may have obscured the effects of the older units transfused.

In Material and Methods, 2.1. Study design and setting: "The PRBC additive solutions used were ...., respectively." It is not clear what respectively refers to. I do not see three different groups described in this paragraph. Were there three hospital settings that used different additive solutions? Please clarify.

In several sections, you refer to PRBC units with a storage age >42 days. Are you referring to frozen deglycerolized units (these are very unusual, used for rare antigens) or is there a solution media in Germany that allows for storage beyond 42 days?

In Discussion: the phrase "In line with these conceptual pathomechanisms of renal injury caused by transfusion of aged PRBCs [34]" is missing a verb.

Author Response

Reviewer #1:

I recommending a few other clinical trials to the references: 

1) - Fergusson DA, Hébert P, Hogan DL, et al. Effect of fresh red blood cell transfusions on clinical outcomes in premature, very low-birth-weight infants: The ARIPI randomized trial. JAMA 2012;308:1443-51.

- Dhabangi A, Ainomugisha B, Cserti-Gazdewich C, et al. Effect of transfusion of Red Blood Cells with longer vs shorter storage duration on elevated blood lactate levels in children with severe anemia. JAMA 2015;314:2514-23.

- Spinella PC, Tucci M, Fergusson DA, Lacroix J, Hébert PC, et al. Effect of Fresh vs Standard-issue Red Blood Cell Transfusions on Multiple Organ Dysfunction Syndrome in Critically Ill Pediatric Patients: A Randomized Clinical Trial. JAMA 2019 Dec 10;322(22):2179-2190. 

- Shah A, Brunskill SJ, Desborough MJ, Doree C, Trivella M, Stanworth SJ. Transfusion of red blood cells stored for shorter versus longer duration for all conditions. Cochrane Database Syst Rev. 2018 Dec 22;12(12):CD010801.

Their incorporation may affect some of the introduction content. For ethical reasons, the clinical trials could not randomize patients to a long-storage RBC transfusion regimen and instead had to compare fresh units to standard units. This may have obscured the effects of the older units transfused.

We appreciate the Reviewer´s comment and have incorporated recommended other clinical trials on page 11 of the Discussion section of the revised version of the manuscript. We have further highlighted the issue that the RCTs could only compare transfusion of fresh PRBCs to transfusion of standard PRBCs in both, the Introduction and Discussion section of the revised version of the manuscript. Because the recommended references focus on pediatric patient populations and the initially cited RCTs and our results were obtained in critically ill adult patients, for the in depth discussion of our results, we focused on the adult ICU population.

2) In Material and Methods, 2.1. Study design and setting: "The PRBC additive solutions used were ...., respectively." It is not clear what respectively refers to. I do not see three different groups described in this paragraph. Were there three hospital settings that used different additive solutions? Please clarify.

During the study period, patients were transfused with leukoreduced, allogeneic PRBCs. However, due to the retrospective setting, robust identification of the additive solutions of all transfused PRBC units was not possible. Therefore, all PRBC additive solutions used during the study period were named: Adsol, PAGGS-M (phosphate-adenine-glucose-guanosine-saline-mannitol), and SAGM (saline-adenine-glucose-mannitol). We have revised the paragraph in the Methods section on page 3 of the revised version of the manuscript accordingly.

3) In several sections, you refer to PRBC units with a storage age >42 days. Are you referring to frozen deglycerolized units (these are very unusual, used for rare antigens) or is there a solution media in Germany that allows for storage beyond 42 days?

Although the majority of manufacturers adhere to the maximum storage of 42 days in the clinical routine, in Germany, PRBCs produced with PAGGS-M additive solution are approved for storage up to 49 days (1). Therefore, to adjust the study cohort to the international standards, patients transfused with PRBCs that were stored for >42 days (n=17) were excluded from the analysis. We have included this information on page 3 in the Methods section of the revised version of the manuscript accordingly.

  1. Flegel WA, Natanson C, Klein HG. Does prolonged storage of red blood cells cause harm?. British Journal of Haematology 2014;165:3–16.

4) In Discussion: the phrase "In line with these conceptual pathomechanisms of renal injury caused by transfusion of aged PRBCs [34]" is missing a verb.

We thank the Reviewer for making us aware of this mistake. The sentence was corrected appropriately in the revised version of the manuscript on page 11.

Reviewer 2 Report

The manuscript by  JA Graw entitled "Age of Red Cells for Transfusion and Outcomes in Patients with ARDS" is of great interest due to the common controversy with the so called RBC storage lesion. Many publications have placed the storage time at 14 days, 32 days, etc. Some manuscripts showing a poor outcome in old blood compared with better outcome in fresher blood.

The authors studied two groups of patients with ARDS who received PRBCs transfusions:  two storage duration periods was analyzed.  Patients transfused with PRBCs of a mean storage age 28 days were compared to patients transfused with PRBCs of a mean storage age >28 days. The primary endpoint was 28-day mortality. Results showed: "patients with ARDS, transfusion of PRBCs with a mean storage age greater than 28 days was not associated with an increase in 28-day mortality. However, transfusion of long-term stored PRBCs was associated with a lower chance for successful weaning from RRT within 28 days after ARDS onset"

Comments

  1. The 28 day cut-off was an arbitrary cut-off point or had some biological basis?
  2. Although there is not a significant difference between the two groups in terms of a 28 day mortality, there is a tendency for better outcome in basically all parameters studied.  There is an improvement to  get free for RRT. Did the authors have data on a longer follow up of these patients?

Author Response

Reviewer #2:

1) The 28 day cut-off was an arbitrary cut-off point or had some biological basis?

During storage, PRBCs undergo a significant number of biochemical and structural alterations with a significant storage-induced damage after 28 days [1, 2]. Therefore, a storage age of 28 days was chosen as a cutoff to group patients into a short-term storage group (PRBC mean storage age ≤28 days) and a long-term storage group (PRBC mean storage age >28 days). This information is now highlighted in the Methods section on page 3 of the revised version of the manuscript.

  1. Donadee C, Raat NJ, Kanias T, Tejero J, Lee JS, Kelley EE, Zhao X, Liu C, Reynolds H, Azarov I et al: Nitric oxide scavenging by red blood cell microparticles and cell-free hemoglobin as a mechanism for the red cell storage lesion. Circulation 2011, 124(4):465-476.
  2. D'Alessandro A, Kriebardis AG, Rinalducci S, Antonelou MH, Hansen KC, Papassideri IS, Zolla L: An update on red blood cell storage lesions, as gleaned through biochemistry and omics technologies. Transfusion 2015, 55(1):205-219.

2) Although there is not a significant difference between the two groups in terms of a 28 day mortality, there is a tendency for better outcome in basically all parameters studied. There is an improvement to get free for RRT. Did the authors have data on a longer follow up of these patients?

We appreciate the Reviewer´s comment. The median observation time was 20 days (IQR, 17-24) in the short-term storage group and 17 days (15-24) in the long-term storage group with respect to the primary endpoint with no difference in censoring between the two groups (P=0.79). Therefore, we were able to use a time frame of 28 days for the endpoints, which is also a commonly accepted time frame for short-term outcome in studies on ICU patients with ARDS. We have included this information in Figure 3 with the respective Figure caption. Because applying a longer follow-up time frame for the studied endpoints does not add value in this retrospective design, the results need to be confirmed in future prospective studies where then a longer follow-up should be applied.